# Characterizing medaka visual features using a high-throughput optomotor response assay

**Risa Suzuki** [1,2], **Jia Zheng Woo** [1¤], **Thomas Thumberger** [1], **Gero Hofmann** [1], **Joachim Wittbrodt** [1]*, **Tinatini Tavhelidse-Suck** [1]*

1 Centre for Organismal Studies (COS), Heidelberg University, Heidelberg, Germany, 2 Heidelberg Biosciences International Graduate School (HBIGS), Heidelberg, Germany

¤ Current address: Department of Biochemistry, Stanford University, Stanford, California, United States of America

* jochen.wittbrodt@cos.uni-heidelberg.de (JW); tinatini.tavhelidse@cos.uni-heidelberg.de (TTS)

**Data Availability Statement:** All relevant data are within the manuscript and its Supporting Information files.

## Abstract

Medaka fish (*Oryzias latipes*) is a powerful model to study genetics underlying the developmental and functional traits of the vertebrate visual system. We established a simple and high-throughput optomotor response (OMR) assay utilizing medaka larvae to study visual functions including visual acuity and contrast sensitivity. Our assay presents multiple adjustable stripes in motion to individual fish in a linear arena. For that the OMR assay employs a tablet display and the *Fish Stripes* software to adjust speed, width, color, and contrast of the stripes. Our results demonstrated that optomotor responses were robustly induced by black and white stripes presented from below in the linear-pool-arena. We detected robust strain specific differences in the OMR when comparing long established medaka inbred strains. We observed an interesting training effect upon the initial exposure of larvae to thick stripes, which allowed them to better respond to narrower stripes. The OMR setup and protocol presented here provide an efficient tool for quantitative phenotype mapping, addressing visual acuity, trainability of cortical neurons, color sensitivity, locomotor response, retinal regeneration and others. Our open-source setup presented here provides a crucial prerequisite for ultimately addressing the genetic basis of those processes.

## Introduction

The teleost medaka (*Oryzias latipes*) is a vertebrate model widely used in basic as well as biomedical research. The advantages of medaka as a vertebrate genetic model system include short generation time, transparent embryos, numerous offspring, a small genome size (about 700 Mb) and easy husbandry [1, 2]. The full repertoire of genome editing tools is available to generate transgenic lines, gene knockout models or human disease models [3–5]. In addition, its high tolerance towards inbreeding enabled the establishment of the first near-isogenic vertebrate population panel [1, 6–9] to explore the genetics underlying a given phenotype [10]. These features make medaka an ideal model for population genomic studies, drug screens and toxicology studies.

**Funding:** This research was funded through the European Research Council (ERC) under the European Union's Horizon 2020 research and innovation program (grant agreement No 810172), https://erc.europa.eu/funding.

**Competing interests:** The authors have declared that no competing interests exist.

Retinal architecture and development are remarkably conserved among all vertebrates. The development, structure and function of the medaka retina has been studied extensively and in great molecular detail [11–13]. For evaluating visual function, behavioral assays have been established as indispensable tools [14, 15]. The optokinetic response (OKR) assay is one of the classical assays where the reflexive eye movements of immobilized animals are analyzed while the eyes are being exposed to alternating black and white stripes placed on a rotating drum [16–21]. In contrast, the optomotor response (OMR) assay evaluates the behavioral locomotor response of animals stimulated by moving black and white stripe patterns. In those assays, visual functions such as spatial resolution or acuity and temporal resolution are evaluated by adjusting stripe width, the speed of stripe movement and the intensity of light exposure [16, 22–26]. OKR and OMR assays have been used in studies to explore the genetic and molecular mechanisms underlying retinal development [23], retinal degenerative disease [27], neural activities [28, 29], as well as animal behavior [30] in various models.

In order to explore the genetics underlying visual sensory related phenotypes a high-throughput method to determine optical features of the individual is indispensable. OKR however has limited throughput due to the immobilization step. Moreover, despite the suitability of OMR for large scale and high-throughput studies given its simplicity, the majority of studies using OMR employ a drum setup with apparent limitations in the throughput.

In the drum-style OMR, a series of alternating black and white stripes rotates around a single animal, in response to which the animal follows the direction of the rotating stripes [23]. One successful instance of using OMR in a high-throughput study was done by Neuhauss et al. [31] where up to a hundred zebrafish larvae were simultaneously placed in a chamber above a flat monitor displaying alternating black and white stripes moving linearly towards one direction. This experimental design enabled screening and identifying zebrafish mutants with visual defects. Yet the setup does not allow the evaluation of the visual features at the level of the individual. The setup is furthermore not designed to eliminate additional stimuli such as the interaction between larvae.

To elucidate the features of individual medaka larvae we established a linear-pool-style OMR assay combining a tablet with full control of all features (stripe width, speed, color and contrast) with a linear-pool-arena with 15 separate lanes to characterize the basic features of fish eyes, such as visual acuity and contrast sensitivity in a highly parallel manner.

## Results

OMR has been utilized for juvenile and adult medaka fish to evaluate their behavior as well as visual functions [23, 30]. In these studies, drum-style OMR which restricts the scalability of the assay, was employed where the fish were placed in a tank inside of a drum showing a moving stripe pattern eliciting a strong behavioral response.

Here, we developed a linear-pool-style OMR setup and protocol for large scale and high-throughput experiments on individual larvae. In the linear-pool-style OMR, the OMR is evoked by stripe motion displayed below the linear pool on a screen. Our open-source *Fish Stripes* software allows full control of stripe parameters (width, speed, color and contrast) (Fig 1A, 1B and S1 Fig). Videos are acquired using a camera (CellCam Centro 200MR) with an infrared (IR) filter and an IR lamp. Data are analysed with the analysis macro (see Materials and methods). With this setup, 15 individual larvae can be tested at a time which are placed individually in each lane of the linear pool (Fig 1C, 1D and S2 Fig).

The larvae were placed in the linear-pool-arena 2 hours prior to the onset of stripe motion with the display showing stationary stripes (acclimation period). In each experiment, stripe motion consists of 4 phases: in phase 1, the stripes are stationary for 5 seconds, in phase 2, the

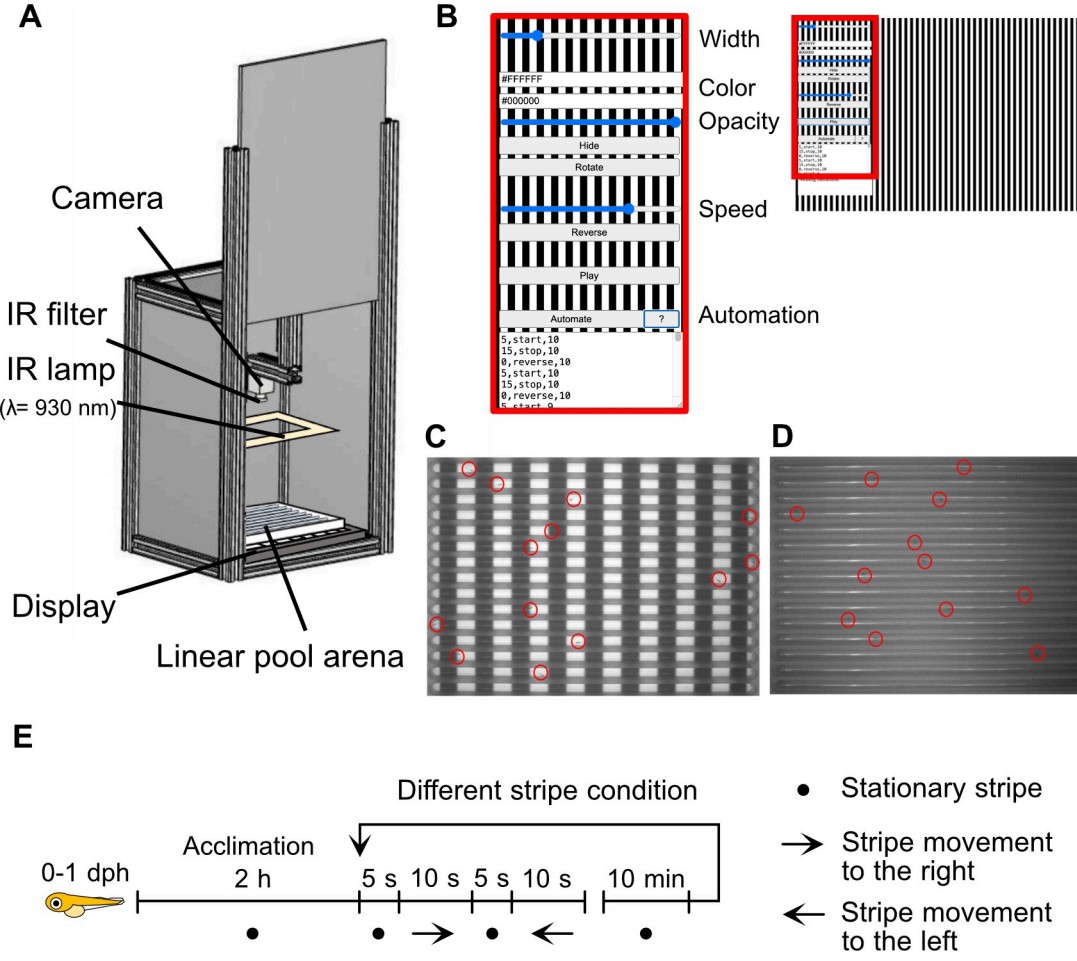

**Fig 1. The linear-pool-style optomotor response (OMR) setup.** (A) Illustration of the linear-pool-style OMR setup: A PVC box containing a display (e.g. tablet) and the 140 mm x 110 mm linear-pool-arena on top. A camera (CellCam Centro 200MR) with an infrared (IR) filter and an IR lamp are used to acquire the videos. The box is closed on all sides to exclude external visual stimuli from the surrounding. (B) Stripe pattern displayed on the tablet screen. Stripe parameters (width in pixels, stripe color 1 and stripe color 2 in hex code, contrast ("opacity") (x 100%) and speed (x 50 pixel/s)) are set and automated using the *Fish Stripes* software. (C, D) Images of the linear-pool-arena placed on top of a tablet screen displaying black and white stripe pattern acquired with a camera without (C) and with IR filter (D). The lanes of the linear-pool-arena are oriented at 90˚ towards the moving stripe pattern. The larvae are marked with red circles. (E) Experimental timeline. 0–1 days post hatch (dph) larvae were acclimatized for 2 hours in the setup while being presented stationary stripes before start of the experiment composed of 5 s stationary stripes, followed by 10 s of stripes moving to the right, 5 s stationary stripes, 10 s of stripes moving to the left. Prior to testing additional stripe conditions, larvae were acclimatized for 10 minutes with respective stationary stripes.

stripes start to move toward right for 10 seconds, in phase 3, the stripes pause for 5 seconds, and in phase 4, the stripes move toward left for 10 seconds. Prior to testing different stripe conditions, larvae were acclimatized for 10 minutes with respective stationary stripes (Fig 1E).

In order to allow enough space for responding, only those larvae with a minimal distance of 27 mm from the end of the lane were considered valid. Those larvae swimming in the direction of stripe motion for at least 20% (27 mm) of the lane length after the initiation of stripe movement were considered responsive. Those larvae that were valid for more than three times out of four exposures to the stripe movement were used for response rate calculation. The response rate was calculated for each larva as the ratio of response count to the count of larva classified valid (S3 Fig).

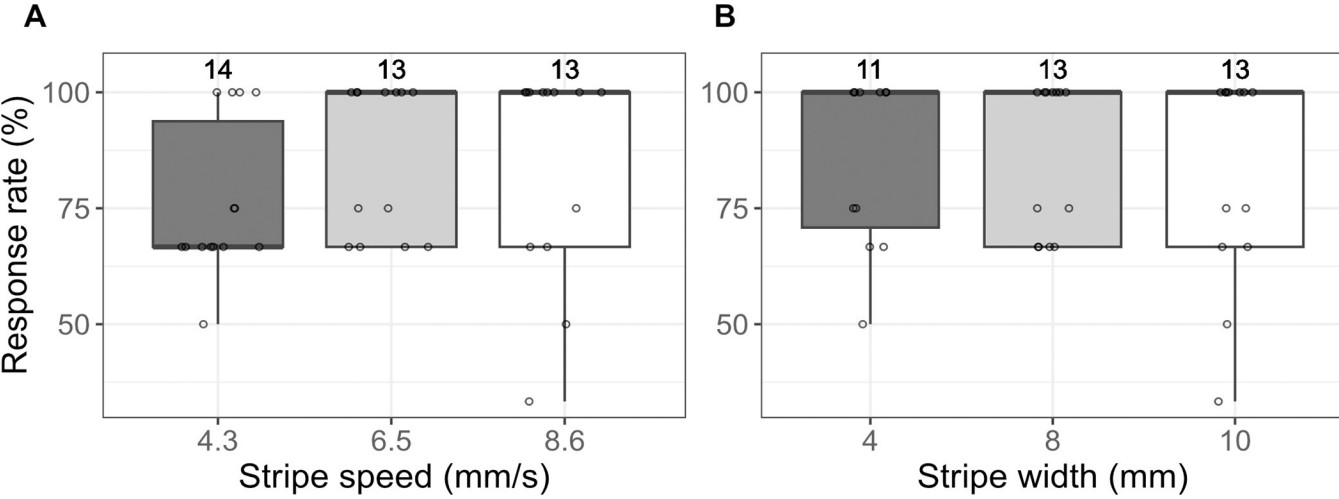

**Fig 2. Medaka Cab strain shows a robust OMR induced by stimulation from below.** (A, B) Response rates of larvae to moving stripes with three different stripe speeds and a constant stripe width of 8 mm (A) or with three different stripe widths and a constant stripe speed of 6.5 mm/s (B). Number above each boxplot represents the number of larvae included for the response rate calculation. Cab strain medaka responded robustly toward black and white stripes at stripe speeds between 4.3 and 8.6 mm/s and stripe widths between 4 and 10 mm. Statistical analysis performed in R, pairwise t-test, Bonferroni corrected.

To assess whether the stimulation from below can induce OMR in freshly hatched Cab strain medaka larvae (0–1 days post hatch (dph), before the initiation of self-feeding [32]), we exposed 15 larvae to black and white 8 mm "thick" stripes of three different speeds: 4.3, 6.5 and 8.6 mm/s (Fig 2A) and stripes of three distinct widths: 4, 8 and 10 mm with a speed of 6.5 mm/s (Fig 2B).

Cab strain medaka showed a robust OMR regardless of stripe width or speed tested (the median response rate was 66.7% for stripe width of 8.0 mm and speed of 4.3 mm/s, and 100% for all other conditions) (Fig 2). All larvae showed a high response rate when the stripes were 8 mm wide with a speed of 6.5 mm/s. Therefore, stripe speed was set to 6.5 mm/s for the following experiments to assess the threshold for stripe width—and hence visual acuity—where fish no longer recognize a "thin" stripe as a stripe and as a consequence no longer show OMR.

In addition to Cab strain medaka, the HdrR strain, a traditional inbred strain originated from Southern Japan, is also commonly used as a wildtype [7]. To test the hypothesis whether visual features are genetically encoded, we used the setup to compare the response between Cab and HdrR strain medaka. 13–15 larvae of each strain were exposed to stripes of 1.4, 1.2, 1.0 and 0.8 mm width respectively (Fig 3). While Cab strain medaka showed a higher response rate towards stripes of 1.4, 1.2 and 1.0 mm width compared to the HdrR strain, the overall response rate in both strains decreased with decreasing stripe width (Cab: median response rate of 66.7% for 1.4 mm, 16.7% for 1.2 mm, 29.2% for 1.0 mm, 0.0% for 0.8 mm, and HdrR: 50.0% for 1.4 mm, 0.0% for 1.2 mm, 0.0% for 1.0 mm, 0.0% for 0.8 mm).

In order to test whether the OMR to different stripe widths is a trainable feature, we extensively exposed 13–15 larvae of each strain to "thick" stripes (8 mm width) for more than 10 times before they were subjected to stripe widths of 1.4, 1.2, 1.0 and 0.8 mm respectively.

In the Cab strain, the response rate toward the thinner stripes increased (median response rate of 66.7% for 1.2 mm and 29.2% for 1.0 mm) but didn't increase for stripe widths of 1.4 mm (66.7%) and 0.8 mm (0.0%) (Fig 3).

Strikingly, after training, HdrR larvae were much more sensitive to thin stripes, showing a significant increase in the response rate to 1.2, 1.0, and 0.8 mm wide stripes (median response rate of 75.0% (1.4 mm), 66.7% (1.2 mm), 50.0% (1.0 mm), and 33.3% (0.8 mm)) (Fig 3).

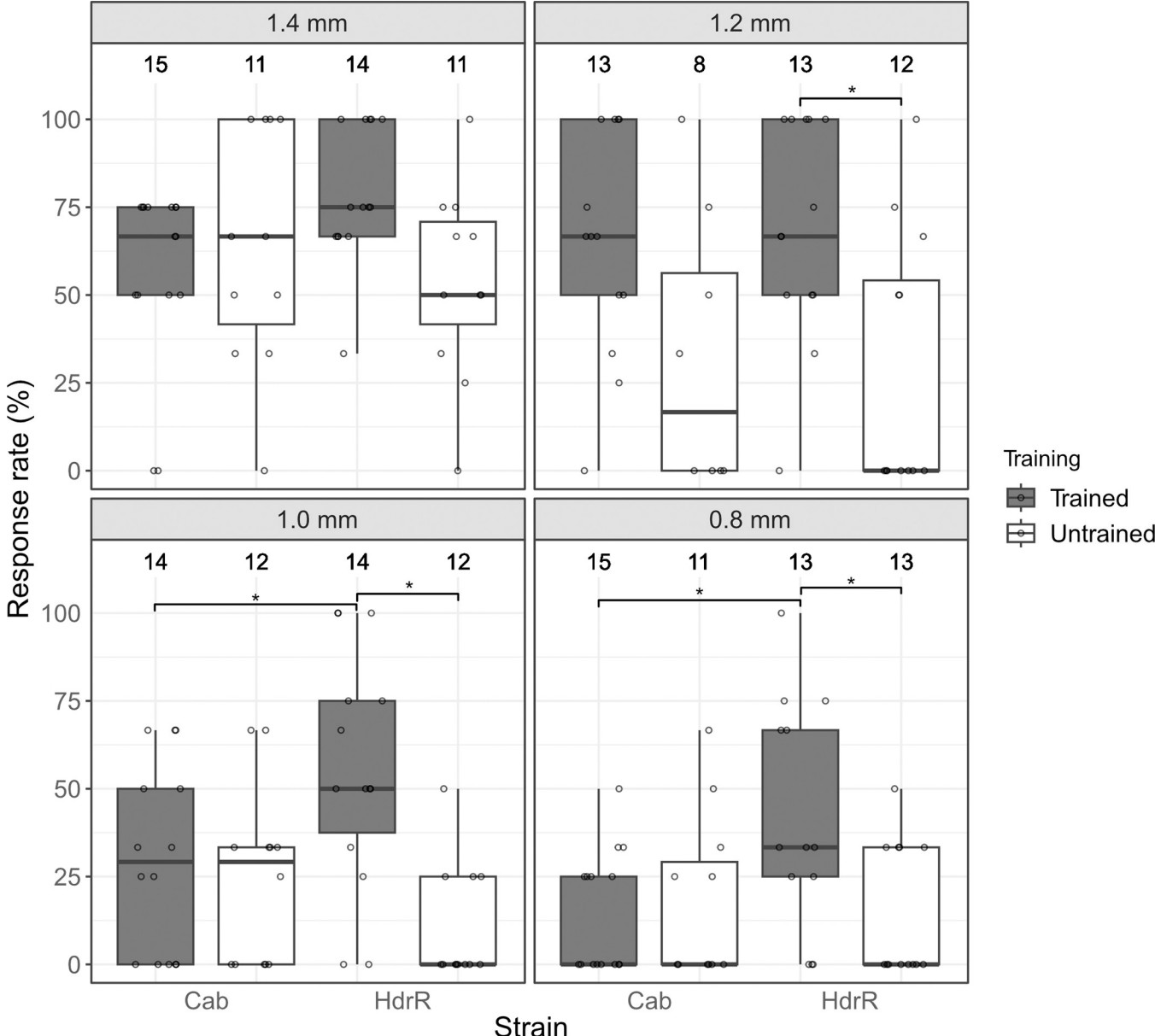

**Fig 3. HdrR strain medaka show higher trainability in their OMR towards narrow stripes than Cab strain medaka.** Response rates of Cab and HdrR strain larvae to moving stripes for each stripe width (1.4, 1.2, 1.0 and 0.8 mm) with (grey) and without (white) prior exposure to thick stripe (8 mm) motion. The response rate gradually decreased with decreasing stripe width both in Cab and HdrR strain medaka. After the training with thick stripes, an increase in response rate was observed both in Cab and HdrR, with a more prominent increase in HdrR. Number above each boxplot represents the number of larvae included for the response rate calculation. * p ≤ 0.05, statistical analysis performed in R, pairwise t-test, Bonferroni corrected.

Interestingly, for stripe widths of 1.0 and 0.8 mm, the trained HdrR strain responded significantly better than the trained Cab strain (Fig 3). These results indicate first, that indeed OMR to different stripe widths is a trainable feature and second, that there is a difference in trainability among different inbred medaka strains.

To assess whether sensitivity to different colors can also be tested using our setup, 15 larvae of each Cab and HdrR strain, as well as of three inbred strains from the Medaka Inbred Kiyosu-Karlsruhe (MIKK) panel [8] were subjected to 8 mm wide black/white, blue/white, green/white or red/white stripes moving at 6.5 mm/s (Fig 4).

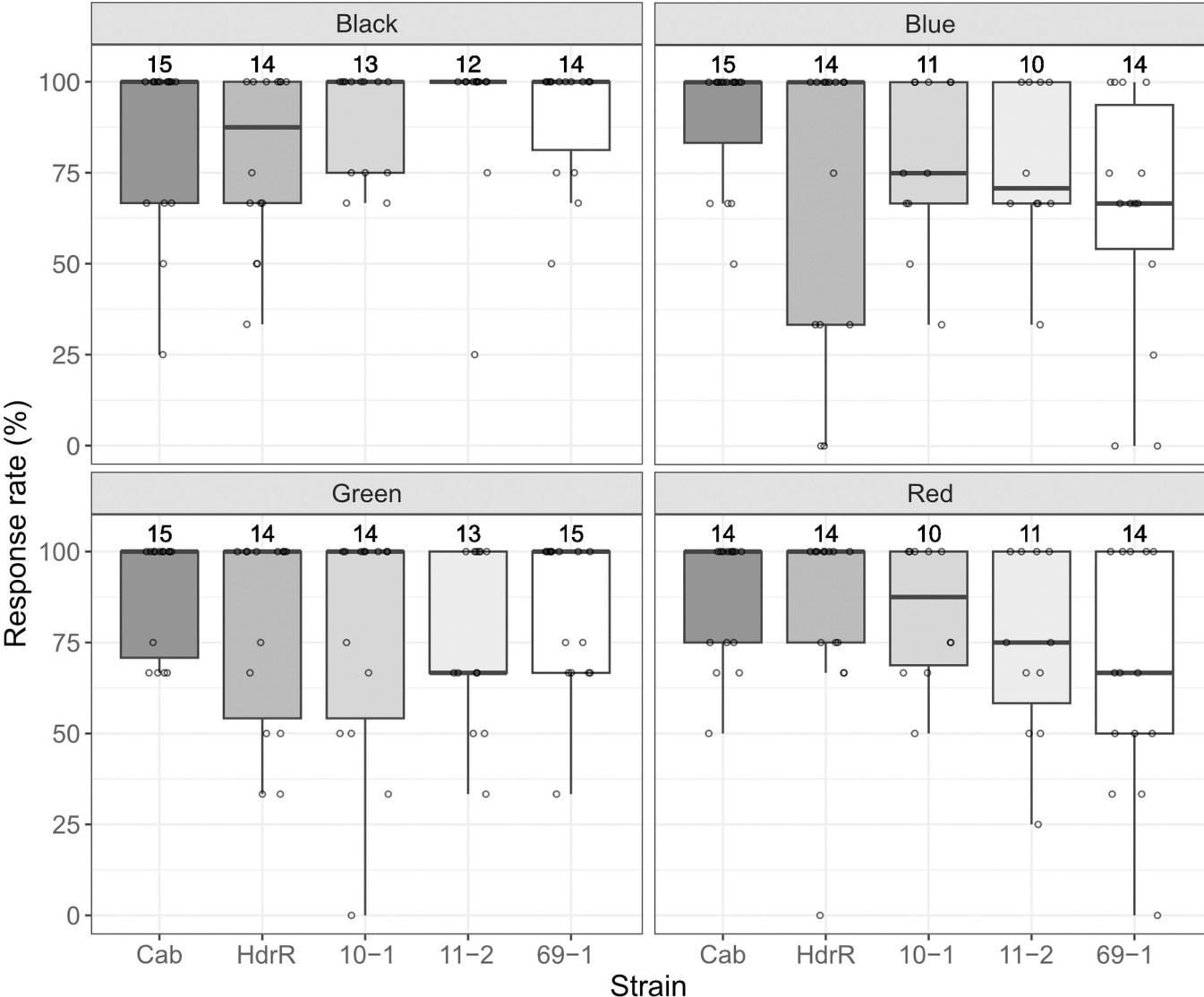

**Fig 4. Various medaka strains show a robust OMR induced by colored stripes.** Response rates of Cab, HdrR, 10–1, 11–2, and 69–1 strain larvae to moving stripes (8 mm wide, moving at 6.5 mm/s) of each color combination (black/white, blue/white, green/white and red/white). Number above each boxplot represents the number of larvae included for the response rate calculation. Statistical analysis performed in R, pairwise t-test, Bonferroni corrected.

Similar to Cab and HdrR strain medaka larvae, larvae of all three inbred strains tested robustly responded to the black and white stripes (10–1, 11–2 and 69–1: median response rate of 100% to black/white stripes). Whereas Cab and HdrR strain medaka larvae robustly responded also to all other color combinations tested, larvae of the three inbred strains tested tended to respond less well to blue/white and red/white color combinations (Cab: median response rate of 100% for all the conditions, and HdrR: 87.5% for black/white and 100% for all the other conditions, 10–1: median response rate of 75.0% for blue/white and 87.5% for red/white, 11–2: 70.8% for blue/white and 75.0% for red/white, 69–1: 66.7% for blue/white and red/white). For green/white stripes, 11–2 responded less well (median response rate of 66.7%) compared to the other strains (10–1 and 69–1: 100%) tested. These results not only show that spectral sensitivity can be tested using our setup, but also that different strains show differential responses to distinct colors.

## Discussion

In this study we developed an easy and high-throughput assay for investigating medaka's visual function using OMR. With our linear-pool-setup employing the open-source *Fish Stripes* software that generates moving stripe patterns and the analysis macro, we provide a full package to easily assess visual features such as visual acuity or color sensitivity for aquatic animals.

In our setup, OMR is robustly induced in 0–1 dph medaka larvae using black and white stripes presented from below, regardless of the stripe speed or width as long as it was thick enough (4–10 mm) and at adequate speed (4.3–8.6 mm/s). So far OMR in medaka has been elicited by presenting stripes rotating around the individual and has preferentially been shown only from 5 dph onwards [23, 24, 30]. In contrast to earlier reports [23], we show a robust OMR in different strains already at 0–1 dph.

This response in earlier stages may be due to the longer acclimation time improving the response rate. With a short acclimation time of less than 30 minutes, Cab strain medaka showed a decrease in response.

Testing visual properties of Cab and HdrR strain medaka by OMR using narrow stripes showed that the response rate depends on the stripe width. Strikingly, we uncover a strain-specific trainability of OMR with HdrR strain medaka larvae being more sensitive to narrow stripes after training. This aligns well with previous findings on strain-specific differences in visual acuity [24].

This trainability is possibly due to the enhancement in retinal computation or the increase in the selectivity of cortical neurons for visual stimuli. The vertebrate retina does not only function as signal transmitter to the brain but also adapts to light and sharpens the image and has its own feedback loop between cells to compute the signal and adjust depending on the stimulus [33]. Possibly, through the training period, contrast gain control and adaptation are acquired by retinal ganglion cells, which enhances the computational ability of the retina itself [34, 35]. Trainability in cognitive function is well studied in mammals as they learn to distinguish task-relevant visual stimuli and the timing to pay attention to the environment, which in return causes a significant change in their early sensory cortical neurons [36]. As the formation and maintenance of attention to visual stimuli has also been reported in fish, the training effect could be explained as due to the increase in the awareness and attention to the environment [37].

Our setup also allows for assessing spectral sensitivity. Color perception is mediated by distinct absorption spectra of different classes of cone-opsins. Medaka possesses 8 cone opsins and 1 rod opsin, which collectively confer sensitivity across the ultraviolet, blue, green, and red regions of the light spectrum [38]. While Cab and HdrR medaka strains exhibit comparable response towards all the colored stripes tested, the three inbred strains from the MIKK panel showed a slightly decreased response. The observed strain specific variations in the response rates to colored stripes may be attributed to potential differences in the expression of the opsin genes.

Here, we established a straightforward, easy-to-recreate and -use, high-throughput OMR-based protocol to study behaviors that are dependent on visual input across medaka and other aquatic species. The 15-lane linear-pool-arena allows parallel assessment of visual parameters at the level of the individual. Thus, this setup provides the basis to address the underlying genetics of various phenotypes not only limited to visual acuity or color sensitivity but also of more complex traits such as retinal regeneration, neuronal selectivity and trainability.

## Materials and methods

### Medaka stocks and maintenance

All medaka strains are maintained in closed stocks at Heidelberg University. Medaka husbandry (permit number 35–9185.64/BH Wittbrodt) was performed according to local animal

welfare standards (Tierschutzgesetz §11, Abs. 1, Nr. 1) and in accordance with European Union animal welfare guidelines [39]. The fish facility is under the supervision of the local representative of the animal welfare agency.

Medaka (*O. latipes*) fish were bred and maintained as previously described [40]. The two isogenic medaka strains used in this study were Cab and HdrR-II1 (strainID: IB178) was supplied by NBRP Medaka (https://shigen.nig.ac.jp/medaka/). All experimental procedures were performed according to the guidelines of the German animal welfare law and approved by the local government (Tierschutzgesetz §11, Abs. 1, Nr. 1, husbandry permit number 35–9185.64/BH Wittbrodt).

Medaka fish larvae used were 0 to 1 day post hatch (stage 40) [41], before the initiation of self-feeding [32]. They were maintained in a constant recirculating system at 28˚C on a 14 hours light/10 hours dark cycle in Embryo Rearing Medium (ERM) (17 mM sodium chloride, 0.4 mM potassium chloride, 0.27 mM calcium chloride dihydrate and 0.66 mM magnesium sulphate heptahydrate at pH 7).

## Linear-pool-style optomotor response (OMR) setup and protocol

The OMR setup is illustrated in Fig 1A–1D. The box is made from extruded aluminum as pillars, PVC sheets as walls, aluminum foil as roof, and stainless steel leveling as feet, all of which were fixed using tapes, and electro-galvanized and passivated steel nuts and bolts. The semi-transparent (70% transmission) linear-pool-arena with a 15-channeled chamber is custom made from plexiglass (PLEXIGLAS® Satinice weiss (snow) WH10 DC). Channels are 5 mm wide, 135 mm long and 5 mm deep.

The linear pool containing one larva per lane was placed on a tablet screen displaying the striped pattern. A camera (CellCam Centro 200MR–Mono Camera) containing a Goyo Optical GM12HR41216MCN lens and an infrared light filter (FI5830-55, Heliopan Filter 5830 | Ø 55 mm, Infrared Filter RG 830 (87C) 830 nm) was mounted above the tablet for video recording. The camera was operated using μManager software (Micro-Manager-2.0.0-gamma1-20210214). Infrared light strips (SECURITY LINE—LED Flex Strip infrared IR 12V IP65 940nm) were placed around the optic to illuminate the arena.

See S1 and S2 Figs for blueprint and technical details.

After 2 hours of acclimation in the linear-pool-arena with stationary stripes, freshly hatched medaka larvae were exposed to stripe motion. One sequence of stripe motion consisted of 4 phases, in phase 1, the stripes are stationary for 5 seconds, in phase 2, the stripes start to move toward right for 10 seconds, then in phase 3, the stripes pause for 5 seconds, and in phase 4, the stripes move toward left for 10 seconds. Prior to testing different stripe conditions, larvae were acclimatized for 10 minutes with respective stationary stripes. Note that the direction of stripe movement is only a perspective of the observer and that there is no systematic preference for a given direction.

To assess whether stimulation from below can induce an OMR in 0–1 dph Cab strain medaka larvae, 15 larvae were exposed to 8 mm "thick" stripes in two consecutive rounds of 4.3, 6.5, and 8.6 mm/s speed. For testing different stripe width, the same 15 larvae were exposed to two consecutive rounds of 4 and 10 mm stripe width with a speed of 6.5 mm/s.

For comparison of OMR between Cab and HdrR medaka strains, 13–15 larvae each were exposed to two consecutive rounds of stripes with 0.8, 1.0, 1.2 and 1.4 mm width and constant speed of 6.5 mm/s (untrained). 15 larvae each were trained by exposure to 11 rounds of stripe motion with 8 mm stripes and constant speed of 6.5 mm/s before exposure to stripes with 0.8, 1.0, 1.2 and 1.4 mm width and constant speed of 6.5 mm/s (trained). Note that the "training" works with any regime/stimulus that triggers a response. Here, thicker stripes were selected as the most robust trigger.

For comparison of OMR toward colored stripes among Cab, HdrR, 10–1, 11–2, and 69–1 strains, 15 larvae each were exposed to two consecutive rounds of stripes with 8 mm width, constant speed of 6.5 mm/s and blue/white, red/white, green/white, or black/white color combinations respectively.

All stripes were presented at 100% contrast. All experiments were carried out at 22 degrees Celsius.

## Generation of the striped pattern

The striped patterns were generated using a JavaScript-based programme, called *Fish Stripes* (https://junshern.github.io/fish-stripes/). The programme allows for user control of stripe width in the unit of pixels, where 1 pixel corresponds to 0.2 mm, speed in the unit of pixel per frame where 50 frames are programmed to be equivalent to approximately one second, colors in hex code, as well as playing, pausing and reversing stripe movement. Timer functions are also available to each of the play, pause and reverse functions for automation purposes.

## Larval tracking and analysis

To track the behavior of larvae, we converted our videos into TIFF files, and analyzed them using Fiji [42]. From the full video, those time frames with stripe motion were extracted. The TIFF stack was processed as follows: first, the first slice of the stack was subtracted from each slice of the stack to remove the background, then median with radius = 6 was produced to extract only the fish position. After that, slices of the stack were color-coded using a Fiji plugin (https://github.com/ekatrukha/ZstackDepthColorCode) and a Z-projection was produced to measure the distance covered by the fish and the initial point of the fish swimming behavior (S3 Fig panel A, see below for macro). For the response rate calculation, those larvae with initial position too close to the lane end (within 20% of the lane length) were considered non-valid and not included for the calculation. Those larvae swimming in the direction of stripe motion for more than 20% of the lane length were considered responsive (S3 Fig panel B). Only those larvae considered valid for at least three out of four stripe movements (two rounds implying two stripe movements in both directions) were included for the response rate calculation. The response rate for each larva was calculated as the ratio of the count of responses within four stripe movements to the count of the larva being considered valid within four stripe movements. Statistical analysis (ANOVA, pairwise t-test, Bonferroni corrected) was performed in RStudio [43].

## Macro for analysis

```
//Select the folder with videos (.tif stacks) to be processed
    showMessage("Select Folder with Videos");
    openDir = getDirectory("Choose a Directory");
    //Select the folder to save the final images
    showMessage("Select Folder to Save");
    saveDir = getDirectory("Choose a Directory");
    //Make a list of.tif stacks in the folder
    list = getFileList(openDir);
    for (i = 0; i<list.length; i++){
    //set batch mode
    setBatchMode(true);
    //Open.tif stack
    open(openDir+list[i]);
    Raw_video = getTitle();
```

```
//Remove any letters after "." in the title
title = substring(Raw_video, 0, indexOf(Raw_video, "."));
//Choose name for saving the first frame of the stack
savename = title + "_first_slice.tif";
//Duplicate the frames where stripe movement lasts for 10 seconds
run("Duplicate. . .", "duplicate range = 0–200");
Dup_video = getImageID();
//Close unnecessary images
close("\\Others");
//Duplicate the first slice of the stack to produce image to subtract
run("Duplicate. . .", "slices = 1");
Dup_image = getImageID();
//Save the first slice for determining valid fish
selectImage(Dup_image);
saveAs("Tiff", saveDir + savename);
//Subtract the first slice from each slice of the stack
imageCalculator("Subtract create 32-bit stack", Dup_video,Dup_image);
//Reduce noise
run("Median. . .", "radius = 6 stack");
run("Invert", "stack");
processed_image = getTitle();
selectWindow(processed_image);
//Close unnecessary images
close("\\Others");
//Run Z projection with color coordinate
//Download plugin from https://github.com/ekatrukha/ZstackDepthColorCode (add
https://sites.imagej.net/Ekatrukha/ to the list of update sites)
run("Z-stack Depth Colorcode 0.0.2", "use = Ice generate = [Colorbar with all 256 LUT col-
ors] output = [Color (RGB)]");
//Run Z Projection with max intensity
selectWindow("Depth_colorcoded_"+processed_image+"");
run("Z Project. . .", "start = 5 projection = [Max Intensity]");
trajectory = getTitle();
//Save the z-projected image
savename_2 = title + "trajectory.tif";
selectWindow(trajectory);
saveAs("Tiff", saveDir + savename_2);
//Close everything
close("*");
}
```

## Data visualization

Data visualization and figure assembly was performed using Fiji [42], ggplot2 [44] in RStudio
[43], Autodesk Inventor 2024 and Affinity Designer 1.10.5.

## Supporting information

**S1 Fig. Technical drawing of the linear-pool-style optomotor response (OMR) setup.**
(TIF)

**S2 Fig. Technical drawing of the linear-pool-arena.**
(TIF)

**S3 Fig. Image processing and analysis scheme.** (A) Tiff stack was processed using Fiji as follows. First slice of the stack was duplicated and subtracted from each slice of the stack to subtract the background. Subsequently, noise reduction using median filter and inversion were performed. Lastly, Depth color code plugin and Z projection were used to visualize larval swimming trajectories. (B) For image analysis, the first slice of the stack was used to classify larvae as either valid or non-valid for response rate calculation. Those larvae located more than 27 mm from the lane end (corresponding to 20% of the lane length) were counted as valid (yellow circles). Those larvae located close to the lane end were considered as non-valid (red circles). Z projection images were used to classify larvae as either responsive or non-responsive. Those larvae swimming in the direction of stripe motion for at least 20% of the lane length were considered as responsive (green circles). Arrow indicates direction of stripe movement.
(TIF)

**S1 File. Classification of individual response.**
(CSV)

**S2 File. Datapoints of Fig 2.**
(CSV)

**S3 File. Datapoints of Fig 3.**
(CSV)

**S4 File. Datapoints of Fig 4.**
(CSV)

## Acknowledgments

We thank all current and previous members of the Wittbrodt and Birney labs for their critical, constructive feedback on the procedure and the manuscript. We thank I. Brettell for his great initial input in setting up the pipeline for fish tracking, image processing and analysis. We thank J. S. Chan for writing the *Fish Stripes* software. We thank N. Grammling for helping equipping the setup and are thankful to M. Majewski, E. Leist, S. Erny and A. Saraceno for excellent fish husbandry.

## Author Contributions

**Conceptualization:** Risa Suzuki, Jia Zheng Woo, Thomas Thumberger, Joachim Wittbrodt, Tinatini Tavhelidse-Suck.

**Data curation:** Risa Suzuki, Thomas Thumberger, Joachim Wittbrodt, Tinatini Tavhelidse-Suck.

**Formal analysis:** Risa Suzuki.

**Funding acquisition:** Joachim Wittbrodt.

**Investigation:** Risa Suzuki, Jia Zheng Woo, Thomas Thumberger, Joachim Wittbrodt, Tinatini Tavhelidse-Suck.

**Methodology:** Risa Suzuki, Jia Zheng Woo, Thomas Thumberger, Gero Hofmann, Joachim Wittbrodt, Tinatini Tavhelidse-Suck.

**Project administration:** Joachim Wittbrodt.

**Supervision:** Joachim Wittbrodt, Tinatini Tavhelidse-Suck.

**Validation:** Risa Suzuki.

**Visualization:** Risa Suzuki, Joachim Wittbrodt, Tinatini Tavhelidse-Suck.

**Writing – original draft:** Risa Suzuki, Joachim Wittbrodt, Tinatini Tavhelidse-Suck.

**Writing – review & editing:** Risa Suzuki, Jia Zheng Woo, Joachim Wittbrodt, Tinatini Tavhelidse-Suck.

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
