## [Decision Letter · Decision Letter 0]

24 Apr 2024

PONE-D-24-12393Characterizing medaka visual features using a high-throughput optomotor response assayPLOS ONE

Dear Dr. Tavhelidse,

Thank you for submitting your manuscript to PLOS ONE. After careful consideration, we feel that it has merit but does not fully meet PLOS ONE’s publication criteria as it currently stands. Therefore, we invite you to submit a revised version of the manuscript that addresses the points raised during the review process. I have added the comments by the two reviewers. The main point is insufficient statistical treatment of the data. Please submit your revised manuscript by Jun 08 2024 11:59PM. If you will need more time than this to complete your revisions, please reply to this message or contact the journal office at plosone@plos.org. Please include the following items when submitting your revised manuscript:A rebuttal letter that responds to each point raised by the academic editor and reviewer(s). You should upload this letter as a separate file labeled 'Response to Reviewers'.A marked-up copy of your manuscript that highlights changes made to the original version. You should upload this as a separate file labeled 'Revised Manuscript with Track Changes'.An unmarked version of your revised paper without tracked changes. You should upload this as a separate file labeled 'Manuscript'.

We look forward to receiving your revised manuscript.

Kind regards,

Stephan

Stephan C.F. Neuhauss, Ph.D.

Academic Editor

PLOS ONE

Journal Requirements:

Reviewers' comments:

Reviewer's Responses to Questions

**Comments to the Author**

1. Is the manuscript technically sound, and do the data support the conclusions?

Reviewer #1: Yes

Reviewer #2: Yes

2. Has the statistical analysis been performed appropriately and rigorously? 

Reviewer #1: No

Reviewer #2: No

3. Have the authors made all data underlying the findings in their manuscript fully available?

Reviewer #1: Yes

Reviewer #2: Yes

4. Is the manuscript presented in an intelligible fashion and written in standard English?

Reviewer #1: Yes

Reviewer #2: Yes

5. Review Comments to the Author

Reviewer #1: Suzuki et al., present a really nice early stage medaka larvae developmental high throughput OMR system. This is a nice study showing data of how sensitive and high throughput this methodology can be. The ability to compare highly inbred medaka strains make it a particular powerful vertebrate genetic model system.

Thus, carrying out this study with two of these strains and showing some differences goes a long way towards highlighting the usability and sensitivity of the OMR. So, I think a key result of the manuscript lies in the differences presented in untrained and trained Cab vs HdrR strains shown in Figure 3. The manuscript is well written and the presented methods would be of great use to the science community.

Main concerns:

1) While the conclusions are reflected by the data, my main concern is the lack of statistical analysis of the data.

Unless I missed it, it seems there were no replicates conducted with different clutches of 15 larvae?

This makes it difficult to judge the robustness of the differences between the strains.

2) In particular, Figure 1 shows differences between left and right responses that are larger than the responses observed between cab and HdrR.

I don't think that this is because vision in the left is wired differently. This might be a result of the experimental design in which the direction of the stripes are always right first.

This could involve a "learning" aspect, where the fish start expecting the second stimulus to be in the opposite direction. This should at least be discussed. If a small supplementary trial with a naive batch, in which one might present:

- left and right

- right and left

- left and left

- right and right

perhaps just at the 6.5 mm/s 8 mm thick stripes.

3) For figure 3, having replicates and showing SEM between these groups would allow for statistical comparison would be really important. For me the robustness and sensitivity of the high throughput method this would highlight is important. This would show whether this approach can indeed distinguish differences in visual performance between the strains, as well as trained vs. untrained - although the difference that training seems to make looks substantial enough to be a significant finding.

Minor points - clarifications please:

- Could the authors add a statement about what stage the visual system is add 0 - 1 dph?

At what stage to medaka need to start feeding (using visual cues)?

- Visual development (as all development) occurs rapidly, so the range in ages 0 - 1 dph, seems quite large.

Are all of the data shown from a single experiment on the same day, same time?

- Is there a reason the "training" was done with the thicker stripes? Would it work with any moving stimuli or is there something particularly specific about the training regime?

- Was there a reason why the direction of the stripes were not randomised?

- Is there a way to revisit the recordings and see, if there was a bias between which way the larvae were facing at the onset of stimulus? I understand that this should be random and thus around 50%, but it might be useful to see, if the 10s stimulus was sufficient to allow larvae facing the opposite way to turn around and start swimming the 27 mm distance required for a positive response.

- I presume all stripes were presented at 100% contrast, could you confirm in the methods please?

- Does the program allow for sinusoidal visual stimuli to avoid edge artefacts?

The figures are nice and clear, looking forward to a revised version.

Reviewer #2: • Statistical evaluation of the data is completely missing. The authors should compare the data between the two fish lines as well as between the different directions.

• The data presented in Figure 3 point to a bias towards the left in the responses. This should be statistically evaluated and discussed.

• Line 192: The strong increase in the response to 1.2 mm stripe width in the Cab strain should be further evaluated in order to determine if this is an outliner or a more general result.

• Line 206: As there is no color testing in the experiments the authors cannot claim their system is working for color sensitivity.

• Figure 2 & 3 & Supp. 4: Standard deviations or standard error of the means, respectively, should be calculated and added to the columns.

• For the comparison of OKR and OMR responses the following article might add some important information: https://doi.org/10.3389/fncir.2021.709048.

6. PLOS authors have the option to publish the peer review history of their article (what does this mean?). If published, this will include your full peer review and any attached files.

Reviewer #1: No

Reviewer #2: No

---

## [Author Response · Author response to Decision Letter 0]

6 Jun 2024

Reviewer #1:

Suzuki et al., present a really nice early stage medaka larvae developmental high throughput OMR system. This is a nice study showing data of how sensitive and high throughput this methodology can be. The ability to compare highly inbred medaka strains make it a particular powerful vertebrate genetic model system.

Thus, carrying out this study with two of these strains and showing some differences goes a long way towards highlighting the usability and sensitivity of the OMR. So, I think a key result of the manuscript lies in the differences presented in untrained and trained Cab vs HdrR strains shown in Figure 3. The manuscript is well written and the presented methods would be of great use to the science community.

Main concerns:

1) While the conclusions are reflected by the data, my main concern is the lack of statistical analysis of the data.

Unless I missed it, it seems there were no replicates conducted with different clutches of 15 larvae?

This makes it difficult to judge the robustness of the differences between the strains.

We thank the reviewer for this comment and have now included the statistical analysis (ANOVA, pairwise t-test, Bonferroni corrected) which we performed in RStudio. We are happily accepting the suggestion to put additional experimental data. We have now added data on three additional inbred strains, as well as different stripe color combinations under standard stripe conditions (new Figure 4). Taken together, this does not only emphasize the robustness of the assay, but also shows that our setup allows for assessing spectral sensitivity.

We have clearly stated these aspects in the revised version of the manuscript and incorporated them in the new Figure 4.

2) In particular, Figure 1 shows differences between left and right responses that are larger than the responses observed between cab and HdrR.

I don't think that this is because vision in the left is wired differently. This might be a result of the experimental design in which the direction of the stripes are always right first.

This could involve a "learning" aspect, where the fish start expecting the second stimulus to be in the opposite direction. This should at least be discussed. If a small supplementary trial with a naive batch, in which one might present:

- left and right

- right and left

- left and left

- right and right

perhaps just at the 6.5 mm/s 8 mm thick stripes.

The reviewer is correct and points out one of the constraints of the assay here. We have presented a consistent direction and order of stripe movement. By first moving the stripes in one direction we collected the larvae at one end of the arena in order to have a starting point for the assay. The larvae perform equally well in both directions (right/left; left/right), since there is no systematic preference in the direction of stripe movement, the direction is only a perspective of the observer. To assay left vs right eyes an alternative assay would require the larvae swimming in circles, which is not possible in the current setup. We have clarified this in the revised version of the manuscript.

To assess “learning” effects, a mechanical way of collection would probably rather be the way to go.

We would also like to clarify that the left and right responses shown in the previous Figure 2 were obtained at stripe widths of 4, 8 and 10 mm, whereas the difference in response between Cab and HdrR strain was observed at a much smaller stripe width between 0.8 - 1.4 mm (Figure 3). In the revised version of the manuscript we compared the response rate of individual larvae to each stripe condition instead of averaging the group response. The response rate was calculated considering four stripe movements (two rounds implying two stripe movements in both directions). In the now new Figure 2 the data are represented in a boxplot including statistical analysis (ANOVA, pairwise t-test, Bonferroni corrected, performed in RStudio), thus making the former representation of responses to left and right stripe movement obsolete.

3) For figure 3, having replicates and showing SEM between these groups would allow for statistical comparison would be really important. For me the robustness and sensitivity of the high throughput method this would highlight is important. This would show whether this approach can indeed distinguish differences in visual performance between the strains, as well as trained vs. untrained - although the difference that training seems to make looks substantial enough to be a significant finding.

We thank the reviewer for this comment and have now included the statistical analysis (ANOVA, pairwise t-test, Bonferroni corrected) which we performed in RStudio. The analysis confirms the initially observed trend and reveals statistical significant differences in response rates between trained and untrained HdrR strain medaka larvae, as well as a significantly better response of trained HdrR strain compared to trained Cab strain for stripe widths of 1.0 mm and 0.8 mm (new Figure 3). We have clearly stated these aspects in the revised version of the manuscript and incorporated them in the revised Figure 3.

Minor points - clarifications please:

- Could the authors add a statement about what stage the visual system is add 0 - 1 dph?

In medaka, retinal differentiation starts around 2.5 days post fertilization (developmental stage 27) (Del Bene et al., 2007, Iwamatsu, 2004) and is completed by 5 days post fertilization, 4 days prior to hatch (developmental stage 34) where all three nuclear layers and the two plexiform layers are well differentiated (Kitambi et al., 2008). Our assay, as well as the study from Furukawa et al. (2002) shows that the visual system is functional at 0 - 1 dph, since the larvae follow the visual cue of stripe movement. Further maturation and thickening of the photoreceptor cell layer (Kitambi et al., 2008), and hence also fine-tuning of the underlying neural networks to continuously improve the visual system occurs throughout later stages and adulthood. In line with this, an improvement of the visual system throughout development has been described in medaka (Carvalho et al., 2002, Furukawa et al., 2002).

At what stage to medaka need to start feeding (using visual cues)?

Medaka start feeding after around 5 days post hatch (depending on temperature and rearing conditions) when the yolk as source of nutrition is no longer sufficient (stage 42) (Iwamatsu, 2004; Watanabe et al., 2023). The first response to visual cues however is escaping predators immediately after hatching, one day before we perform the assays.

- Visual development (as all development) occurs rapidly, so the range in ages 0 - 1 dph, seems quite large.

Are all of the data shown from a single experiment on the same day, same time?

The reviewer ist correct - in case visual development would not be completed at this stage, the range in ages 0 - 1 dph is quite large. However, since development of the visual system is complete at this stage (Kitambi et al., 2008), this range is not too large. For further clarification we would like to point out a key difference between medaka and zebrafish development: while zebrafish hatches at 3 dpf (Kimmel et al., 1995) with its nervous system not yet fully developed, medaka hatches at 10 dpf (Iwamatsu, 2004) with a fully developed nervous and visual system (Furutani-Seiki, Wittbrodt, 2004).

All data are shown from larvae of comparable age in an experiment at comparable time of the day.

- Is there a reason the "training" was done with the thicker stripes? Would it work with any moving stimuli or is there something particularly specific about the training regime?

The “training” works with any regime/stimulus that triggers a response. We selected the most robust trigger which was the thicker stripes and state this also in the revised version of the manuscript.

- Was there a reason why the direction of the stripes were not randomised?

There was no reason why the direction of stripes was not randomised. We have presented a consistent direction and order of stripe movement. By first moving the stripes in one direction we collected the larvae at one end of the arena in order to have a starting point for the assay. The larvae perform equally well in both directions (right/left; left/right), since there is no systematic preference in the direction of stripe movement, the direction is only a perspective of the observer.

- Is there a way to revisit the recordings and see, if there was a bias between which way the larvae were facing at the onset of stimulus? I understand that this should be random and thus around 50%, but it might be useful to see, if the 10s stimulus was sufficient to allow larvae facing the opposite way to turn around and start swimming the 27 mm distance required for a positive response.

Indeed this would be interesting to see, however since the videos were recorded using an infrared lamp and a camera with an infrared filter, only the heavily pigmented parts of the larval bodies can be traced, which are only the eyes.

- I presume all stripes were presented at 100% contrast, could you confirm in the methods please?

We thank the reviewer for pointing this out and have added the respective information (“all stripes were presented at 100% contrast”) in the methods part of the revised manuscript (Line 332).

- Does the program allow for sinusoidal visual stimuli to avoid edge artefacts?

The program does not allow for sinusoidal visual stimuli yet. We consider implementing this in the future in a new version of the program.

The figures are nice and clear, looking forward to a revised version.

Reviewer #2:

• Statistical evaluation of the data is completely missing. The authors should compare the data between the two fish lines as well as between the different directions.

We thank the reviewer for this comment and have included now statistical analysis of the data (ANOVA, pairwise t-test, Bonferroni corrected), revealing significant differences between trained and untrained HdrR exposed to 1.2, 1.0 and 0.8 mm wide stripes, as well as between trained Cab and HdrR subjected to 1.0 and 0.8 mm wide stripes. We have clearly stated these aspects in the revised version of the manuscript and incorporated them in the revised Figure 3.

• The data presented in Figure 3 point to a bias towards the left in the responses. This should be statistically evaluated and discussed.

We thank the reviewer for helping us to resolve this important point. We have presented a consistent direction and order of stripe movement. By first moving the stripes in one direction we collected the larvae at one end of the arena in order to have a starting point for the assay. The larvae perform equally well in both directions (right/left; left/right), since there is no systematic preference in the direction of stripe movement, the direction is only a perspective of the observer. Instead of calculating an average response rate, the data are now represented in boxplots including statistical analysis (ANOVA, pairwise t-test, Bonferroni corrected, performed in RStudio), thus making the former representation of responses to left and right stripe movement obsolete. We have clearly stated these aspects in the revised version of the manuscript and incorporated them in the revised Figure 3.

• Line 192: The strong increase in the response to 1.2 mm stripe width in the Cab strain should be further evaluated in order to determine if this is an outliner or a more general result.

The observed trend of an increase in the response to 1.2 mm wide stripes after training in the Cab strain is indeed statistically not significant. However, the trend is apparent for 1.2 mm wide stripes. No increase in response could be seen for 1.4 mm, 1.0 mm and 0.8 mm wide stripes. Strikingly, for the HdrR strain, the response rate to 1.2, 1.0 and 0.8 mm wide stripes was significantly increased after the training. We have clearly stated these aspects in the revised version of the manuscript and incorporated them in the revised Figure 3.

• Line 206: As there is no color testing in the experiments the authors cannot claim their system is working for color sensitivity.

The reviewer is correct. We have now included a new Figure 4 with data on different stripe color combinations (black/white, blue/white, green/white and red/white) under standard stripe conditions for Cab and HdrR strains, as well as for three additional inbred strains. Even though statistically not significant, there was an apparent trend in Cab and HdrR strains which robustly responded to all color combinations, whereas the three additional inbred strains tested responded less well to blue/white and red/white color combinations. We have clearly stated these aspects in the revised version of the manuscript and incorporated them in the new Figure 4.

• Figure 2 & 3 & Supp. 4: Standard deviations or standard error of the means, respectively, should be calculated and added to the columns.

We thank the reviewer for this important comment and have included now the statistical analysis (ANOVA, pairwise t-test, Bonferroni corrected, performed in RStudio) for Figures 2, 3 and new Figure 4. The data is now represented in boxplots. We have removed Supplementary Figure 4 (S4 Fig), since these data are now included in the revised Figure 3.

• For the comparison of OKR and OMR responses the following article might add some important information: https://doi.org/10.3389/fncir.2021.709048.

We thank the reviewer for pointing out this highly relevant reference to us which we have referred to now in Line 66.

---

## [Decision Letter · Decision Letter 1]

18 Jun 2024

Characterizing medaka visual features using a high-throughput optomotor response assay

PONE-D-24-12393R1

Dear Dr. Tavhelidse,

We’re pleased to inform you that your manuscript has been judged scientifically suitable for publication and will be formally accepted for publication once it meets all outstanding technical requirements.

Kind regards,

Stephan C.F. Neuhauss, Ph.D.

Academic Editor

PLOS ONE

Additional Editor Comments (optional):

Reviewers' comments:

Reviewer's Responses to Questions

**Comments to the Author**

1. If the authors have adequately addressed your comments raised in a previous round of review and you feel that this manuscript is now acceptable for publication, you may indicate that here to bypass the “Comments to the Author” section, enter your conflict of interest statement in the “Confidential to Editor” section, and submit your "Accept" recommendation.

Reviewer #1: All comments have been addressed

Reviewer #2: All comments have been addressed

2. Is the manuscript technically sound, and do the data support the conclusions?

Reviewer #1: Yes

Reviewer #2: Yes

3. Has the statistical analysis been performed appropriately and rigorously? 

Reviewer #1: Yes

Reviewer #2: Yes

4. Have the authors made all data underlying the findings in their manuscript fully available?

Reviewer #1: Yes

Reviewer #2: Yes

5. Is the manuscript presented in an intelligible fashion and written in standard English?

Reviewer #1: Yes

Reviewer #2: Yes

6. Review Comments to the Author

Reviewer #1: Dear authors, thank you very much for addressing all of the points raised so thoroughly.

The extra colour data and the presentation of each datapoint with the relevant box plot have greatly improved the manuscript.

Simplification of Figure 2 is also much clearer now.

I believe this manuscript to present novel data of interest, well analysed, presented and discussed.

Reviewer #2: (No Response)

7. PLOS authors have the option to publish the peer review history of their article (what does this mean?). If published, this will include your full peer review and any attached files.

Reviewer #1: No

Reviewer #2: **Yes: **Volker Enzmann

---

## [Editor Report · Acceptance letter]

21 Jun 2024

PONE-D-24-12393R1 

PLOS ONE

Dear Dr. Tavhelidse-Suck, 

I'm pleased to inform you that your manuscript has been deemed suitable for publication in PLOS ONE. Congratulations! Your manuscript is now being handed over to our production team.

Kind regards, 

on behalf of

Dr. Stephan C.F. Neuhauss 

Academic Editor

PLOS ONE